

# Spatial and Temporal Variability in Baseflow in the Mattole River Headwaters, California, USA

Nathan Queener[1], Andrew P. Stubblefield[2]

[1]Mattole Salmon Group, Petrolia, CA, 95558 USA
[2]Department of Forestry and Wildland Resources, Humboldt State University, Arcata CA, 95521 USA

*Correspondence to*: Nathan Queener (nathan@mattolesalmon.org)

**Abstract.** Increases in human population, water use, and climate change have the potential to increase water stress and scarcity particularly in ecosystems with pronounced seasonality in precipitation, yet our understanding of the landscape features that control baseflows remains limited. Repeated synoptic measurements of streamflow in small streams (basin area <6 km$^2$) in coastal Northern California were used to characterize variability in baseflow and correlations of streamflow with basin characteristics. A continuous streamgage was used as an index gage to calculate exceedance flows and to compare tributary flows measured on multiple dates. At 72-96% exceedance flows tributary yields ranged from 0.23 to 0.00 mm day$^{-1}$. Unit-area yields varied widely, and this variation increased as flows declined at most sites. In nested basins, downstream declines in both discharge and unit-area yield were common. Basins with greater summer flow and a slower baseflow recession had steeper slopes, higher elevations, less flat ground and narrower valleys, more dissected and strongly convergent topography, and more precipitation. The difference in water yield among basins was much greater than the difference in precipitation, likely resulting from varying basin water inputs, storage capacity, and routing. The positive correlation between basin steepness and flow is attributable to the thickness of the weathered bedrock layer in water storage, and more rapid bedrock weathering in steeper basins with higher rates of uplift resulting in greater basin storage capacity. Results show that basins in a small geographic area (<85 km$^2$) and with fairly similar geology, vegetation, and topography may generate widely differing baseflow. Streams with naturally low baseflows are particularly susceptible to water diversion. Low-gradient streams essential for coho salmon rearing may be particularly susceptible to climate change or water diversions that reduce streamflow.

**Keywords**: *baseflow, groundwater recharge, drainage density, coho salmon, bedrock aquifer*





# 1 Introduction

The frequency, magnitude, and duration of minimum streamflow are primary drivers of the structure and function of riverine ecosystems (Rolls et al. 2012). Declining streamflow affects a suite of physical and chemical factors that alter

biological interactions within aquatic ecosystems, especially in climates with highly seasonal precipitation regimes (Gasith and Resh 1999, Lake 2003). With increases in human population, water use, and climate change all exacerbating water stress and scarcity in coming decades (Gasith and Resh 1999, Moyle et al. 2013) there is a need to improve understanding of the processes that regulate dry season baseflows in order to inform action to preserve or enhance flows.

In the Mediterranean climate of northern California and southern Oregon, interest in streamflow is driven by

concern for salmonids (Moyle et al. 2013, National Marine Fisheries Service 2014) and household and agricultural use (Deitch et al. 2009, Schremmer 2014). Climate change models predict increases in temperature (Bell et al. 2004, Cayan et al. 2008), and evapotranspiration and the seasonal disconnection between peak water inputs and peak water demand (Deitch et al. 2009, Grantham et al. 2010). Streams in this region have already experienced declining minimum flows and steepening recession curves (Asarian and Walker 2016, Madej 2011, Sawaske and Freyberg 2014).

This study focused on small catchments (basin area <6 km$^2$) in the Mattole River watershed in northern California with the goal of investigating streamflow variability among tributary streams with relatively similar drainage areas, and seemingly similar vegetation, geology, climate and land-use history (Klein 2009). Physiographic metrics were selected from those shown in the literature to be influential.

The specific mechanisms that exert the greatest control on baseflow timing and magnitude vary substantially

depending on scale, climatic, topographic, and geologic setting and individual studies are sometimes conflicting (Price 2011, Smakhatin 2001, Tetzlaff et al. 2009). The magnitude and timing of precipitation is an independent control on streamflow generation (Boughton et al. 2009, Price et al. 2011). In Mediterranean climates, late spring precipitation may be critical (Hunter et al. 2005, Klein 2009, Reid and Lewis 2011, Reid 2012). The depth of the hydrologically active regolith within a basin exerts a primary control on watershed unit/area storage capacity (Buttle et al. 2004, Sayama et al. 2011, Smakhatin

2001). Conditions at the soil/bedrock or saprolite/bedrock contact may be influential in determining storage capacity. Permeable bedrock, or the depth of weathered bedrock or saprolite may constitute the primary reservoir of storage for baseflow generation (Liu et al. 2013, Price 2011, Rempe and Dietrich 2014, Salve et al. 2012, Sayama et al. 2011). While soil depths are typically assumed to be greater on lower gradient slopes and in valley fill (Price 2011), studies of the depth of the hydrologically active weathered bedrock zones in argillite and granite have found increasing depth with higher slope

position (Holbrook et al. 2014, Rempe and Dietrich 2014, Salve et al. 2012).

The length of time water takes to traverse a catchment and move between hillslope, riparian, and stream components depends on both the properties of the regolith and the topography of the basin. The hydraulic conductivity of the





regolith and the hydraulic gradient determines transit time through a particular subsurface medium. However less conductive layers may actually decrease transit times by increasing the proportion of surface runoff (Tetzlaff et al. 2009).

Researchers have found a correlation between gentler slopes and increased transit times and/or baseflows (McGuire et al. 2005, Price 2011). In other studies, steeper slopes and greater relief have been positively correlated with increased baseflows, slower recessions, and/or transit time, typically due to covariance between regolith characteristics, precipitation or watershed storage capacity (Price et al. 2011, Sánchez-Murillo et al. 2014, Tetzlaff et al. 2009).

Transit time is also influenced by flowpaths. Surface water moves much more rapidly than subsurface, and in general the presence of more and deeper flowpaths increases travel time (Wondzell et al. 2010). Greater drainage density effectively results in more surface flow relative to subsurface due to channels intersecting hillslope storage, and has been found to have a negative relationship with baseflow magnitude (Moore and Wondzell 2005, Price 2011, Price et al. 2011). Overland flow path length has been found to have a positive correlation with transit time and baseflow (McGuire et al. 2005, Price 2011).

Upslope accumulated area (UAA), the hillslope area draining to each stream pixel in the channel network, predicts the proportion of time a hillslope is hydrologically connected to the stream system (Jencso et al. 2009). However as flows decline, this relationship weakens. A decreasing proportion of a basin is hydrologically connected to the stream (Jencso et al. 2009) and the influence of heterogeneous bedrock flow paths increases (Payn et al. 2012).

At the hillslope-riparian and riparian-stream interfaces the hydraulic gradient and the hydraulic conductivity of the substrate determine the degree that runoff from upland areas is transmitted or deflected (Bernal and Sabater 2012, Hunter et al. 2005, McGlynn and Seibert 2003, Moore and Wondzell 2005, Richard et al. 2013, Storey et al. 2003). When the riparian water table is low, hillslope water will recharge the riparian water table prior to contributing to streamflow (Bernal and Sabater 2012, Butturini et al. 2002). With a high riparian water table, hillslope water may be forced into flowpaths parallel to the riparian-hillslope margin, or emerge as surface water. However, during the dry season these same conditions are associated with transmission loss, and can make stream water available for transpiration by riparian vegetation (Sophocleus 2002, Wondzell et al. 2010).

Losses of water from a basin during the baseflow period consist of evaporation, transpiration from vegetation, and seepage to deep groundwater (Smakhatin 2001). Of these, transpiration is of most relevance to baseflows in Pacific Northwest forests, often exceeding streamflow in magnitude during the dry season (Reid 2012). Many studies have noted an increase in baseflows following timber harvest due to reduced transpirative demand (Hicks et al. 1991, Keppeler 1998, Moore and Wondzell 2005, Reid 2012). However, harvested basins show a decrease in baseflow as vegetation becomes re-established and transpiration increases (Reid 2012).

The timing and magnitude of transpiration is also influenced by vapor pressure deficit (Emanuel et al. 2014), stand age, density, sapwood basal area, and species composition. Younger, denser, forests in the Pacific Northwest appear to transpire more moisture than older stands and trees (Moore et al. 2004, Stubblefield et al. 2012). Alder (Alnus) species transpire more water than conifers (Hicks et al. 1991, Moore et al. 2004). Broadleaf evergreens and conifers draw water from



different portions of the regolith (Link et al. 2014, Zwieniecki and Newton 1996). Douglas fir water use peaks in spring and decreases throughout the dry season in response to water stress, while Madrone (Arbutus) transpiration reaches a maximum in late summer (Link et al. 2014).

In some catchments water retained in the soil and available for use by vegetation, and water that contributes to streamflow may constitute nearly separate pools of water (Brooks et al. 2010). Water that contributes to baseflow may move through larger pores and preferential flow pathways deep into the saprolite, out of the reach of vegetation except where regolith depths are relatively shallow (Liu et al. 2013, Salve et al. 2012). Transpiration by riparian vegetation may be less limited by water availability and thus have a pronounced effect on baseflows (Bernal and Sabater 2012, Bren 1997, Hicks et al. 1991, Moore et al. 2011, Reid 2012, Richard et al. 2013).

Patterns in baseflow recession and surface flow persistence are an expression of the internal structure of a watershed, and the balance of water inputs, storage capacity, transit time, and loss. Conversely, a lack of correlation between baseflow magnitude or recession and basin characteristics may indicate that poorly quantified aspects of catchment structure, or feedbacks and covariance among processes exert a dominant control on baseflow. This study aimed to increase our understanding of the patterns and processes of stream drying through a specific case study in the Mattole River headwaters. Objectives were to describe temporal and spatial variability in baseflow and determine if correlations among baseflow and landscape characteristics provide insight into the dominant mechanisms controlling baseflow magnitude in the study streams.

## 2 Materials and Methods

### 2.1 Study Site

The study took place in coastal northern California, USA (Fig. 1) within the southern subbasin of the Mattole River watershed (777 km$^2$) The basins (Table 1, Fig. 1) were chosen for accessibility, lack of human water use, and to span the likely range of dry season streamflow in the Mattole headwaters (Klein 2009 and 2012). Only Anderson, Thompson, and Ancestor Creek basins had any residences, and presumably, consumptive use of surface water.

The climate is Mediterranean, with annual precipitation averaging 190-240 cm (Coates et al. 2002). Nearly all of this precipitation falls as rain from November-May. Streamflow varies seasonally and between years, with Mattole River mean monthly flows at the Petrolia USGS gaging station downstream (drainage area 660 km$^2$) ranging from less than 0.6 to over 226 m$^3$s$^{-1}$. Mean annual streamflow at Petrolia is 2020 mm yr$^{-1}$.

Vegetation in the region is mixed hardwood/conifer forest, dominated by Douglas fir (P. menziesii), coast redwood (S. sempervirens), and tanoak (L. densiflorus). Riparian tree species are predominately red alder (A. rubra), with a minor component of Oregon ash (F. latifolia). Over 95% of the area has been harvested in the last 60 years, and forest composition is dominated by relatively young and dense stands. Predominant land uses are timber management, rural subdivision, and small-scale agriculture, including marijuana cultivation. Approximately 86% of the southern subbasin is privately owned, and the population of the area was 206 in the 2000 census (Downie et al. 2003), 2.5 people km$^{-2}$.





The underlying geology is predominantly sandstone and argillite, classified as the Coastal Belt of the Franciscan Complex (Davenport et al. 2002). When compared to the remainder of the Mattole watershed, the southern subbasin exhibits relatively low rates of mass wasting. Overall relief is also the lowest in the watershed (Davenport et al. 2002). The mainstem Mattole and most larger tributaries are contained in semi-confined channels, incised within broad valley bottoms composed

of strath terraces (terraces created by fluvial incision into bedrock) covered by a thin mantle of Holocene or Pleistocene alluvium (Davenport et al. 2002). Most second-order and larger streams have relatively low gradients (<3% slope) relative to those in the remainder of the watershed. This extensive lower-gradient stream network and lower susceptibility to mass wasting are primary reasons that the southern subbasin contains much of the suitable over-summering habitat for juvenile coho salmon in the Mattole River watershed (Mattole River and Range Partnership 2011).

While many of the streams within the study area have channels partially confined by strath terraces due to their geologic history, reductions in instream wood loading due to intensive timber harvest and removal of wood due to concerns about fish passage has likely resulted in contemporary channel degradation through much of the stream network, and reductions in channel-floodplain connectivity (Mattole River and Range Partnership 2011, Wooster 2000).

**2.2 Methods**

The study approach was to measure discharge within tributary streams of the Mattole River watershed during the dry season, and then calculate a suite of physiographic characteristics using GIS for each basin. Correlations between the magnitude of catchment specific discharge and basin character could then be explored. Because baseflow measurements were made on multiple dates over multiple dry seasons, a flow index approach was used to develop specific flow metrics for

comparison in the correlation analysis.

Volumetric streamflow measurements were conducted with containers, a stopwatch, and a means of channeling flow such as clay or a silicone baking sheet. Where volumetric measurements were not feasible, an electromagnetic velocity meter (Marsh-McBirney 2000) and the area-velocity method were used. With volumetric flow, at least three and up to 15 measurements were made at any one occasion, and the mean of these measurements reported as the discharge for that site

and time. Calculated standard deviations ranged from 0-22% of discharge, with a mean and median of 5%.

To compare water yield among basins of varying size, flow values were converted to specific discharge ($Q_{sp}$) by dividing by basin drainage area, and then expressing the result as unit area yield (mm day$^{-1}$). To compare streamflow and the linear extent of dry channel across multiple years and dates, daily flow values from the U.S. Geological Survey gaging station on the mainstem Mattole at Ettersburg (#11468900) were used as an index gage, and a surrogate for time. The

Ettersburg stream gage is 20 km downstream from the confluence of the Mattole River and Van Arken Creek, the downstream-most tributary in this study. The Mattole River drainage area at the site is 182 km$^2$.





Flow duration curves and flow exceedance probabilities were calculated for the Ettersburg gage, and the flow exceedance at Ettersburg on a specific date assumed to approximate the exceedance for the flow at a specific tributary measured on the same date. The flow exceedance curve was calculated according to Searcy (1959):

$$P = 100 * (\frac{m}{(n+1)})$$

Where P is the exceedance probability, m is the ranking from highest to lowest of all daily mean flows for the period of record, and n is the total number of daily mean flows. The period of record began in June of 2001. Streamflow and specific discharge from paired upstream and downstream sites were plotted against flows at the Ettersburg gage to explore patterns in unit/area yield within catchments as baseflows declined through the dry season. Flow exceedance probabilities may be understood as the percent of days in a year in which flows would be expected to exceed the flow value. For example,

a $Q_{76}$ value of 425 l s$^{-1}$ on the main stem at Ettersberg, would be exceeded 76% of the time, while a $Q_{96}$ value of 150 l s$^{-1}$ would be expected to occur more rarely, with higher flows expected 96% of the time.

The flow metrics chosen: $Q_{76}$, $Q_{85}$, and $Q_{96}$, encompass the widest span of Ettersburg Q for which measurements were taken from a critical mass of tributary sites (Table 2). For the sites at which flows were not measured on dates when Ettersburg daily flows equalled the three values above, discharge was estimated by fitting a regression line to higher and

lower tributary flows plotted against their corresponding Ettersburg Q. The regression equation was then used to calculate tributary Q for the desired Ettersburg Q. This interpolation was only made within the range of measured tributary values. Only the tributary Q measurements nearest the desired value on the Ettersburg hydrograph were used to generate the predictive equation for some sites, to avoid fitting a straight line across an apparent inflection point in the recession curve.

Correlation coefficients were calculated between three streamflow metrics (Table 2) and basin characteristics

selected based on their potential influence on baseflow volume and timing (Jencso et al. 2010, Liu et al. 2013, McGuire et al. 2005, Price 2011, Price et al. 2011, Smakhatin 2001, Tetzlaff et al. 2009). All basin characteristics shown in Table 3 were calculated in the geographic information systems program QGIS (QGIS Development Team 2013), using modules available in the System for an Automated Geographical Analysis (SAGA) processing toolbox contained within QGIS (Böhner et al. 2008).

The 10m Digital Elevation Models (DEMs) used to derive topographic variables were obtained from the USDA Natural Resources Conservation Service Geospatial Data Gateway (http://datagateway.nrcs.usda.gov). DEMs were stitched to create continuous coverage of the area of interest, and pre-processed by filling sinks and using the multi-direction lee filter to remove irregularities at the tile seams. Flow paths and the stream network were delineated using the multiple triangular flow direction algorithm (Seibert and McGlynn 2007), with a minimum drainage area of 0.015 km$^2$ (1.5 ha) for channel

initiation. This threshold was chosen based on limited field mapping of channel heads within the study basins.

Study basin polygons were delineated using the SAGA watershed basin tool, and then inspected and corrected for errors. In addition to the DEMs, other sources of information used in calculating basin variables were vegetation composition data from CALVEG (CDF 2005), surficial geology as interpreted from aerial photos (CGS 2003, Davenport et



al. 2002), and stream channel network characteristics from the NOAA Fisheries Intrinsic Potential (IP) data (Agrawal et al. 2005).

Precipitation and temperature data used were from the PRISM Climate Group (http://prism.oregonstate.edu). PRISM uses factors including elevation, coastal proximity, and aspect to interpolate event-based climatic data from a network of stations to a continuous grid (Daly et al. 2008). Because the spatial resolution of the publicly available grid data (2 x 2 km for 30 year normal data, and 0.8 x 0.8 km for monthly data) exceeded the size of some of the study basins, grids were re-sampled in QGIS to a finer resolution using bilinear interpolation.

The distribution of most calculated physiographic values was non-normal. Consequently mean values for basin variables were used only when distributions were approximately normal. The 25th, 50th, 75th, and maximum values were calculated for basin variables that were not proportions. If these values were highly correlated (r>0.90) only one variable was presented.

All statistics were completed in the program R (R Development Core Team. 2011. R: A language and environment for statistical computing. R Foundation for Statistical Computing, Vienna, Austria). The non-parametric Spearman's correlation coefficient *rho*, was used to assess the relationship between the landscape variables and the flow metrics, due to the non-normal distributions of most basin variables. Distributions of the flow metrics, in particular, tended to be heavily positively skewed and remained so even after applying transformations.

Principal components analysis (PCA) was conducted with the calculated physiographic variables in Table 3 to better understand differences among basins, using Hubert's PCA, a method developed for non-parametric data sets with outliers and skewed distributions (Todorov and Filzmoser 2009). Data were not transformed. PCA was done on two subsets of variables, those relating to weather/climate, aspect, and vegetation, and all variables except for weather/climate metrics.

## 3 Results

### 3.1 Temporal and Spatial Variability in Streamflow

Large differences in low flow unit area water yield were found between tributary subbasins. 143 streamflow measurements were conducted during the low flow seasons of 2011-2013, when daily mean discharge at the Ettersburg gaging station downstream on the Mattole River ranged from 510 to 150 l s$^{-1}$ (72% and 96% exceedance flows, respectively) (USGS 2014).

In 2011, a greater number of flow measurements were taken over a broader span of time and Ettersburg discharges (13 August to 2 November and 510-224 l s$^{-1}$) than in 2012. Measurements were also completed following the first rainfall events of the season, which occurred in the last week of September and first week of October, and more than doubled discharge at the Ettersburg gage.

In 2012 measurements spanned less than a month of the low-flow period, but were completed at lower flows. The first measureable autumn rainfall did not occur until 13 October, although multiple episodes of cooler temperature and



increased humidity interrupted the recession of Ettersburg flows in late August, September, and beginning on October 5th. Four additional tributary measurements were completed in 2013.

Measured tributary flows ranged from $5\,l\,s^{-1}$ to $0\,l\,s^{-1}$. Maximum $Q_{sp}$ recorded was $0.271\,mm\,day^{-1}$, at Hulse Fork of Ancestor Creek on 13 August 2011. This was one of only two streams, along with Danny's Creek, where $Q_{sp}$ at the tributary measurement site exceeded $Q_{sp}$ at Ettersburg. Most tributary unit-area yields were far less than the simultaneous $Q_{sp}$ at Ettersburg. The median ratio of tributary $Q_{sp}$/ Ettersburg $Q_{sp}$ for all measurements was 0.136. Tributary $Q_{sp}$ declined relative to Ettersburg $Q_{sp}$ through the flow recession at all tributary sites except for two, Hulse Fork and Upper Anderson Creek, which showed increased unit-area yield relative to the Ettersburg gage. Discharge data from a total of 17 sites were available for Ettersburg discharges of 272 and $150\,l\,s^{-1}$, calculated as 85% and 96% exceedance flows, although some of these values were extrapolated from flow measurements taken at slightly higher and lower flows at the Ettersburg gage.

The distribution of both discharge and specific discharge values from these 17 sites was strongly right-skewed, with many very low values, especially $Q_{sp}$ values. Maximum, median, and minimum Q values at Q85 were 2.109, 0.534, and $0.184\,l\,s^{-1}$, and at Q96 were 0.543, 0.071, and $0.000\,l\,s^{-1}$. $Q_{sp}$ was 0.1998, 0.0194, and 0.0108 mm day-1 at Q85, and 0.1376, 0.0024, and 0.0000 mm day-1 at Q96. The distribution of $Q_{sp}$ values appeared to be much less skewed at greater discharges. While measurements were only taken from nine sites at an Ettersburg Q of $425\,l\,s^{-1}$, the distribution of these values is nearly normal, compared to the right skew of the areal yields form the same sites at lower flows.

The relative difference in Q and $Q_{sp}$ among sites was much greater at $Q_{96}$ than at $Q_{85}$. At $Q_{85}$ the maximum measured Q at Main stem Ancestor Creek was 12 times the minimum Q measured on that date, at site Lower North Fork of the Lost River. Maximum unit area yield at site Hulse Fork was 18.5 times the minimum $Q_{sp}$, at Lower Main Stem Lost River. Maximum $Q_{sp}$ at $Q_{96}$ was also measured at Hulse Fork, and was 530 times higher than the minimum non-zero value, at site Upper South Fork of the Lost River. The highest Q value, from Upper Anderson Creek was 96 times the lowest non-zero Q from the same date, measured at Lower South Fork Lost River. The drainage areas of Hulse Fork and Upper Anderson Creek are smallest and third-smallest, respectively, among the basins.

Declines in specific discharge from $Q_{85}$ to $Q_{96}$ ranged from 0.0108 to 0.0680 mm day-1, with relative declines of 31 to 100%. The absolute decline in flow was strongly positively correlated with a site's $Q_{85}$ (Spearman's *rho*=0.93, p=<0.01), with an inverse and less strong relationship between the relative decline in flow and $Q_{85}$ (Spearman's *rho*=-0.69, p=0.002). Unit-area yield at sites with a higher $Q_{85}$ decreased more in absolute terms through the recession, but the relative decrease in $Q_{sp}$ from $Q_{85}$ to $Q_{96}$ was greater at sites with lower initial specific discharge.

## 3.2 Declining Downstream Yields

Downstream declines in both discharge and unit-area yield were common. All downstream sites had lower Q and $Q_{sp}$ values than the closest upstream site on at least one occasion. The occurrence and magnitude of declining downstream yield generally increased with decreasing flow at the Ettersburg gage. At the lowest flows encountered, all downstream sites had lower Q and $Q_{sp}$ than adjacent upstream sites. At Ettersburg flows of 181 and $212\,l\,s^{-1}$, two sites, Main Stem Ancestor



Creek and Upper Lost River showed slight gains in Q from upstream. The only positive downstream $Q_{sp}$ at these flows was also at Upper Lost River, but only in relation to Lower South Fork Lost River.

Lower Baker Creek, Upper Lost River, Main Stem Lost River and Lower South Fork Lost River were the only sites where downstream $Q_{sp}$ was greater than upstream yield within the range of flows measured. Greater downstream Q was
observed at least once at all sites except for three, Lower Anderson Creek, Mid Lost River, and Lower North Fork Lost River, although at most sites any observed increase in Q was very small.

At Lower Anderson Creek, Mid Anderson Creek, Mid Lost River, and Lower North Fork Lost River the pattern of decreasing downstream flow with declining Q at Ettersburg was strongly linear. Some irregularities in this relationship at the other sites appears to stem from measurements taken in 2011 following rainfall events on 25 September and the first week of
October. At Lower Baker Creek, the three measurements taken at higher Ettersburg flows that show a very slight gain in downstream flow were taken on 25 September, 2 October, and 2 November. It is possible that Upper Baker Creek responded more rapidly or strongly to these rainfall events, altering the upstream/downstream relationship that existed on recessional flows. Lower Helen Barnum Creek and Lower South Fork Lost River also show evidence of an altered upstream/downstream relationship following rainfall events, with an apparently greater increase in flow at the upstream site.
The relationship of Upper Lost River to the upstream sites Lower South Fork Lost River and Lower North Fork Lost River is more variable than any of the other pairs. Variability in flow measurements may have caused some of the scatter, as there were multiple dates with an SD of >10%. However, the dominant trend is still greater downstream loss as flows decline, and declining $Q_{sp}$ relative to Lower North Fork Lost River. Specific discharge increased relative to Lower South Fork Lost River, although this is mainly attributable to the extremely steep decline in flow at Lower South Fork Lost
River.

## 3.3 Correlation of Basin Characteristics and Low Flow Metrics

While the strength of correlations varied, in general basins with more summer flow and slower flow recession had steeper slopes, higher elevations, less flat ground and narrower valleys, more dissected and strongly convergent topography, and more precipitation. All *rho* values are presented in Tables 4-7, below.
Flows were not strongly correlated with any measure of basin aspect, although there was a weak positive correlation with south-facing aspect (Table 4). Basins with lower gradient stream networks tended to have lower specific discharge, and showed larger declines from Q85 to Q96. Drainage density, often considered an indicator of how efficiently a catchment sheds water, was unexpectedly positively correlated with higher flows at Q76 and Q85. The proportion of basin mantled by quaternary alluvium— a general co-occurrence with lower gradient stream networks—was strongly negatively correlated
with all flow metrics (Table 4).

Negative correlations between baseflow magnitude and total stream length, basin area, basin perimeter, and the maximum distance from drainage divide to outlet (MaxFPL) are all related to the phenomenon of unit area yield declining downstream and with increasing drainage area in the study basins. These correlations were significant for all flow metrics





except for $Q_{76}$. When maximum flowpath length was scaled by dividing by basin area (max.fpl.area), no significant relationship with flow metrics were observed.

Drainages with wider valley floors, and a greater proportion of valley floor to basin area had lower baseflows, and a larger proportional decline in flow from $Q_{85}$ to $Q_{96}$. These correlations were uniformly negative, and strongest with $Q_{85}$,
followed by $Q_{96}$ (Table 4). Steeper, higher elevation basins yielded more baseflow. Median elevation, relative basin relief, median slope, and the median gradient of un-channelized flowpaths were all positively correlated with flow metrics. These steeper basins are also more highly dissected with more convergent topography, and also have higher drainage density. Upslope accumulated area and sub-catchment area, measures of internal catchment structure and the degree of dissection, were negatively correlated with flow magnitude. Median values of the Topographic Wetness Index (TWI), usually indicative
of the potential for saturated soil conditions, were negatively correlated with all flow metrics, particularly so with $Q_{85}$ and $Q_{96}$ (Table 4).

Tree canopy density, intended as a potential surrogate for transpirative demand, was strongly positively correlated with $Q_{76}$ and $Q_{85}$. This is likely a spurious correlation, but may be an indication of some other difference in forest structure not captured in the data available. The proportion of a basin comprised of hardwood forest type was positively correlated
with all flow metrics, while streams with less baseflow generally had more coniferous forest. With the exception of canopy density, none of the vegetation metrics showed particularly strong correlations with the flow metrics.

Parcel density, intended as a coarse way of quantifying human water use, was positively correlated with baseflow magnitude (Table 4). While this is most likely a spurious correlation, it is further evidence that the observed patterns in baseflow in these streams are likely not driven by human water use.

Most monthly mean precipitation totals were significantly positively correlated with $Q_{76}$, $Q_{85}$ and $Q_{96}$. The distribution of July precipitation across the study area was notably different than the other months, with higher totals in the SE of the study area, versus the W and SW for other months, and predominately uncorrelated with the flow metrics (Table 5).

Maximum summer temperatures were negatively correlated with flow, June and August temperatures especially so.
Mean summer temperatures showed the opposite relationship, with significant positive correlations with both July and August values (Table 5). In general, the PRISM data showed maximum temperatures increasing from the SW to the NE across the study area, while mean temperatures increased from E to W. Presumably these opposing patterns are broadly attributable to greater temperature fluctuations further from the coast, with higher daytime temperatures and cooler overnight low temperatures.

The correlations between May precipitation values from 2011 and 2012 and $Q_{85}$ and $Q_{96}$ were positive and statistically significant, as were June precipitation values and $Q_{76}$ and $Q_{85}$ (Table 6). As with basin normal precipitation (Table 5), July values from 2011 and 2012 were not significantly correlated with any flow metrics (Table 6). September precipitation in 2011 was strongly positively correlated with $Q_{85}$.



April values from 2011 were negatively correlated with $Q_{76}$ and $Q_{85}$, while the correlation with $Q_{96}$ (which used 2012 precipitation totals) was strongly positive (Table 6). These opposing results may indicate that April precipitation has little effect on the baseflow magnitude, and these are spurious correlations. Overall, it appears that May, June, August, and September precipitation amounts are positively correlated with baseflow magnitude.

**3.4 Variability in Basin Characteristics**

Principal components analysis was run on two subsets of the variables used to describe basin characteristics. The first run, on the 47 variables not describing weather or climate, provided insight into the topographic characteristics of the basins. The first principal component accounted for 63% of the total variance in the data, and primarily separated basins based on valley width, the gradient of the channel network, and the slope and elevation of catchments (Table 7, Fig. 2).

These are primarily the landscape variables that were most highly correlated with $Q_{sp}$ values. PC2, which explained a much smaller proportion of total variance, was most heavily loaded for basin aspect, forest composition, flowpath lengths, and basin perimeter and length of the stream network. Most variables with strong loadings in Component 2 were not highly correlated with $Q_{sp}$ values. Sub-drainages from the same stream tended to cluster together, which is not surprising given the overlap among nested basins. In general, the drainages with higher discharges are in the upper left-hand corner of Fig. 2,

based on their steeper slopes, higher drainage densities, higher elevations, predominant south-facing aspect, longer overland and entire basin flowpaths. Yew Creek, which had similarly higher flow yield to the Anderson and Ancestor sites, is notably removed from all other sites, based on north-facing aspect and the highest proportion of conifers among all basins, traits not shared by other higher flow sites.

Plotting components 2 and 3 shows a similar clustering of sub-drainages as in the plot of components 1 and 2, but

separates the Anderson and Ancestor sites based on differences in riparian area calculated from the DEM relative to basin area and stream length, stream gradient, basin relief, and the extent of 1st order streams (Fig. 2). Component 2 appears to sort most downstream to upstream sub-drainages from left to right. This appears to be based primarily on the greater basin perimeter and total stream lengths in the larger sub-drainage, and the greater median un-channeled flowpath length (i.e. longer hillslope) and total relative flow distance (from ridgetop to outlet scaled by basin area) in the smaller sub-drainages.

PCA using weather/climate, aspect, and vegetation variables also clustered sub-drainages together, and produced a plot remarkably similar to a mirror image of the basins' geographical location (Fig. 1 and 2). All Lost River and Helen Barnum sites, in addition to Lower Baker Creek and the Mouth of Van Arken Creek, were on the far left side of PC1, based on greater precipitation in April 2011, and higher average maximum June and August temperatures. Positive to negative distribution of sites along the component 2 axis closely approximates their north to south location, and was based on higher

average April and October precipitation, and higher maximum July temperatures further north. Southern sites had higher July precipitation totals, larger trees, and more hardwood-dominated forest cover. Maximum June and August temperatures had strong negative loadings on component 1 and strong positive on component 2, indicating the highest estimated maximum temperatures in the northeastern portion of the study region (Table 7).





## 4 Discussion

The geographically proximate catchments examined in this study share similar topography, geology, and vegetation, but yield widely varying quantities of dry season streamflow, with specific discharge spanning two orders of magnitude at the driest conditions. We will discuss some possible mechanistic explanations for the phenomena observed, uncertainties in the methods employed and data used, and implications for efforts to increase dry season streamflow and improve aquatic habitat.

### 4.1 Temporal and Spatial Variability in Flow

We observed a high degree of both spatial and temporal variability in streamflow and unit-area yield. Relative differences in $Q_{sp}$ among the basins, as well as the skew of the distribution of flow values, appeared to increase as conditions became drier, with a variation in unit area yield of three orders of magnitude at the driest conditions. This increasing variability in yield among basins as flows decline has been observed elsewhere (Payn et al. 2012, Shaman et al. 2004), and attributed to the lessening role of topography and greater role of subsurface structure in determining flow volume as flows decline (Payn et al. 2012, Shaw 2015).

Recession rates slowed as the season progressed, eventually reaching very low flows with gradually declining recessional slopes. An inflection in the recession rates was observed at a $Q_{sp}$ of around 0.01 mm day-1 for sites with enough flow measurements to adequately characterize it. A shift in the recession rate has been interpreted to indicate the depletion of a particular "store" of groundwater from within the basin (Bart and Hope 2014, Shaw 2015). Once a faster-draining groundwater store is depleted, recession rates decrease. Streamflow measurements to define this inflection point would improve understanding of recession rates and the role of hillslope versus riparian sources of baseflow.

### 4.2 Basin Characteristics and Baseflows

Correlation analyses showed strong positive correlations between $Q_{sp}$ and steeper channel networks and hillslopes, higher basin elevation, greater drainage density and a more dendritic stream network, more precipitation, and higher mean temperatures. Significant negative correlations were observed with metrics describing wider stream valleys, a lower gradient channel network, less dissected hillslopes with longer overland flow paths, and maximum summer temperatures. The correlation results suggest that precipitation inputs, the partitioning of storage and flow paths, and losses to evapotranspiration may all exert an important influence on baseflow magnitude in the study streams.

The positive correlation between precipitation and $Q_{sp}$ was not surprising, and has been documented in other studies (Boughton et al. 2009, Price et al. 2011). However, the differences in magnitude of precipitation among the study basins were much less than the difference in magnitude of $Q_{sp}$. Differences in monthly accumulated precipitation among the basins were all less than 25%, while at $Q_{76}$, $Q_{85}$, and $Q_{96}$ the highest yielding basins had $Q_{sp}$ of four times, 18 times, and >530 times



the $Q_{sp}$ of the lowest yielding basins. This suggests that the precipitation data do not reflect the range of actual basin precipitation, or that other factors mediate the rate of movement of rainwater through the basins.

Because flow measurements were used from multiple years, we primarily used 30-year basin normal precipitation on a monthly time-scale. Flow measurements from a single year would allow the use of more precise precipitation data from preceding years. The spatial resolution of the precipitation data was coarse (2 x 2 km). Re-sampling to finer resolution data may not have captured fine-scale variation in precipitation. Elevation and coastal proximity are factors in the interpolation that produces PRISM output (Daly et al. 2008), but rapid decreases in precipitation inland may bias the estimate low. Rainfall data come exclusively from inland stations to the east. However precipitation data from a private weather station in the Thompson Creek drainage (C. Thompson, pers. comm., 2014) closely matched spring and summer totals calculated from PRISM data for study watersheds in 2011 and 2012. These totals support the conclusion that PRISM data do not drastically underestimate rainfall in the seaward study basins, and that difference in unit-area discharge among the basins are much greater than differences in rainfall.

The correlation of metrics describing basin steepness and dissection with baseflow seems counterintuitive. Hydrologic theory generally holds that smaller, steeper basins with high drainage density will have a flashier hydrograph and faster recession (Curran et al. 2012). Although some studies have found positive correlations between basin slope or flow path gradient and baseflow (Sánchez-Murillo et al. 2014, Tetzlaff et al. 2009), these correlations appear to be a result of drastic differences in lithology, with steeper basins having greater water-holding capacity due to different rock or soil types than flatter basins.

The correlation observed in this study between steeper basins and greater baseflows is likely a result of greater water storage capacity in the steeper basins, even though the geologic parent material appears to be fairly uniform across the study area. Recent studies in the South Fork of the Eel River, at a site with nearly identical geology (argillite of the Franciscan Formation), have found that the weathered bedrock layer is the primary source of summer baseflow, and in most places on the landscape has much greater potential for water storage than the soil column, due to its much greater depth (up to tens of meters thick) (Salve et al. 2012). Differential rates of weathering driven by higher rates of groundwater flux on steeper terrain and more frequent repeated wetting and drying of fresh bedrock on slopes may explain thickening of the weathered layer upslope (Rempe and Dietrich 2014). Increasing saprolite thickness upslope has also been documented in granitics in the Sierra Nevada (Holbrook et al. 2014).

Bedrock weathering rates are also increased by uplift (Rempe and Dietrich 2014), and the general trend of decreasing uplift rates moving inland in the region (Merritts et al. 1994) may provide an additional explanation for less storage capacity in the inland and easternmost basins. A positive correlation between bedrock weathering rates and slope creates a hydrologically active layer of greater depth in steeper basins with narrower valley bottoms.

It is more difficult to find a mechanistic hydrologic explanation for the positive correlations between baseflow and drainage density, and negative correlations between baseflow and upslope accumulated area (UAA), sub-catchment area (SCA), and un-channelized flow path length. With greater drainage density more channels intersect subsurface stores of



water and more efficiently carry water out of the basin. This metric has been consistently negatively correlated with baseflow magnitude (Moore and Wondzell 2005, Price 2011, Price et al. 2011). Higher UAA, SCA, and flow path length describe a watershed with longer un-dissected hillslopes and lower channel density. Jencso et al. (2009) found evidence for a UAA threshold size above which hillslopes contributed dry-season flow to stream channels. The inverse relationships found in this study must be due to heterogeneous subsurface conditions or issues with channel network delineation.

In delineating a stream network from a DEM, we used a drainage area threshold of 0.015 km$^2$ for channel initiation, based on limited field reconnaissance. Use of a single value initiation threshold is common (Jencso et al. 2009, McGuire et al. 2005, Tetzlaff et al. 2009), and mapping channel heads over a very large area is probably impractical in most studies, but this assumption of landscape homogeneity is problematic. In a study of the distribution of channel heads in similar terrain in the nearby Elk River watershed in Humboldt County, Buffleben (2011) found the drainage area for channel initiation to vary by over an order of magnitude among 45 channel heads within three tributary streams. Many variables used in analyses of basin characteristics and streamflow are calculated using the derived channel network. The assumption of a set channel initiation threshold across basins may lead to systematic errors in characterization of catchments.

With equivalent hydrologic input, basins with less subsurface storage capacity should have a smaller drainage area at which channel initiation occurs, since surface flow will initiate with less accumulated precipitation than in basins with greater subsurface storage capacity. Within the study area, it is possible that lower flow basins, with less storage capacity, may actually have a smaller threshold for channel initiation than higher flow basins, and therefore a higher than calculated drainage density. It is also possible that the actual extent of the channel network has little effect on baseflow magnitude. If surface topography has a lessening influence on flow magnitude as conditions become drier (Payn et al. 2012), greater subsurface storage capacity may be a more important factor than channel configuration and extent during the dry season.

Correlations between baseflow and metrics that describe wider valley floors and riparian areas were consistently negative. In the study area, some near stream areas are low enough in elevation to be functional floodplains (Fig. 3a) while other valley floors are perched terraces or strath terraces, which are rarely, if ever wetted by streamflow (Fig. 3b). It appears that in many cases the 10 m resolution of the DEM obscured these differences and classified a large portion of the valley floor, which is actually much higher, as within one meter of the stream channel elevation.

For example, in the Lost River drainage much of the main channel in the Upper South Fork site is highly connected to the floodplain (Fig. 3a). Downstream of the site the channel becomes more incised (Fig. 3b), and most of the stream length downstream from the North Fork/South Fork confluence is confined with a limited floodplain and frequent bedrock exposures (strath terraces). However, based on the DEM, the width of the "riparian" within one meter of the channel elevation increases steadily downstream to nearly 100 m near the stream mouth (Fig. 4). Based on field inspections it appears that much of this area is well over one meter above the channel and would only be inundated in very large and infrequent storm events. These different types of near-stream features that were all classified as riparian for this analysis have different hydrologic interactions with the stream and likely vary in substrate composition as well. A more precise delineation of these features might show a different relationship between their occurrence and the magnitude of baseflow.





The negative correlation between basin maximum temperature and baseflow is logical and likely related to higher temperatures causing increased evapotranspiration rates. Dry season transpiration in the Mattole has been estimated at 0.5 mm day$^{-1}$ (Stubblefield et al. 2012), with estimates from the adjacent Eel River basin at 1-2 mm day$^{-1}$ (Link et al. 2014). Both these figures greatly exceed all $Q_{sp}$ values measured in this study. Basins further from the coast generally had higher estimated maximum temperatures. As with precipitation, the difference among maximum temperatures in the study basins was slight, just 5%, much less than the observed differences in $Q_{sp}$.

Despite transpiration's dominant role in the dry-season water flux correlations between variables describing vegetation character and $Q_{sp}$ were weak, or the inverse of expected. Basins with greater canopy density and a greater proportion of broadleaf evergreens had greater specific discharge. Increased canopy density should result in greater transpiration, and hardwoods in the region appear to use more dry season water than do Douglas fir (Link et al. 2014, Zwieniecki and Newton 1996). The coarse nature of the vegetation data may have led to a lack of correlation. Alternatively, where vegetation density is high, water use may still be limited relative to in a less dense stand due to a lack of available solar energy to drive ET (Emanuel et al. 2014). Differences in ET rates may also have been accentuated by un-quantified differences in fog, cloud cover, and humidity among the basins, all of which can effect streamflow. (Sawaske and Freyberg 2015).

### 4.3 Downstream Declines in Yield

In catchments with multiple flow measurement sites there was a strong pattern of declining $Q_{sp}$ and discharge at downstream sites. This could arise from increased transmission loss, or decrease in lateral inputs from hillslopes or riparian areas. Most of the reaches where downstream loss in Q was observed did not appear to have the morphologic features – deep alluvium in an unconfined valley or a contact between different lithologies– that typically contribute to the presence of a strongly losing reach (Larned et al. 2011, Konrad 2006).

While it was impossible to accurately quantify loss rates with no measurements of lateral inflow, loss seemed to be evenly distributed. Transpiration by riparian vegetation may be greater in the downstream portion of a basin due to warmer temperatures and greater irradiance in open valleys (Emanuel et al. 2014, Salve et al. 2012). Also, streamside hillslope gradients were generally less with wider valleys in their lower reaches (e.g. Ancestor Creek in Figure 1). Given the correlation between steeper slopes and baseflow, there may be a lack of water input in these lower reaches. The prevalence of declining downstream yields supports the idea that large portions of these basins contribute little or no water to streamflow during low flow periods.

### 4.4 Valley Fill: Source or Sink?

The negative correlation between baseflow magnitude and valley width, and the ubiquity of downstream decreases in unit/area yield seem to indicate that valley bottoms function primarily as a water sink in dry conditions. Soil survey data





also suggest that coarse textured valley fill has less water holding capacity than hillslope soils in the study area (Natural Resources Conservation Service, United States Department of Agriculture. 2012).

One method to enhance baseflow is to add large woody debris, in an attempt to store more water and slow its' movement through the valley fill. Given the observed importance of hillslope sources in maintaining late-season flow, this approach may have little effect in portions of channel with little hillslope inflow. A more spatially explicit look at the relationships between hillslope and valley characteristics, and losing or gaining reaches might help identify landscape features which contribute to baseflow.

The observed correlation between higher baseflow and steeper catchments does not bode well for the persistence of coho salmon populations if climate change leads to further reductions in dry-season flow. Coho primarily rear in low-gradient streams because they require low-velocity habitat to ride out winter stormflows (National Marine Fisheries Service 2014). Decreasing baseflows may further constrict coho habitat quality and availability across the landscape.

## 5 Conclusions

Greater water storage capacity in steeper hillslopes due to greater depths of hydrologically active bedrock is likely the dominant control on the observed differences in dry season $Q_{sp}$ for small catchments in the Mattole River drainage of Northern California. Differences in precipitation and temperature among basins likely accentuate the variability in $Q_{sp}$.

These results indicate that extreme variability in summer baseflow can occur independent of diversions and consumptive water use. Conversely, a single instream diversion could very easily capture the entirety of the streamflow, potentially for months at a time. Protecting habitat for aquatic species in basins with naturally higher baseflow per unit area is particularly important in the face of the uncertain effects of climate change on precipitation patterns and changes in streamflow.

Estimated transpiration rates are substantially greater than measured unit/area yield in all basins in this study. Forests within the region are also likely denser and more water hungry than they were historically, due to past timber harvest practices and the vacation of the historical fire regime, estimated at 6-25 yr return interval (Lorimer et al. 2009). These facts make forest thinning or prescribed burning seem like possible approaches to streamflow enhancement. However, summer transpiration, especially by Douglas fir, appears to be largely water-limited in the region (Link et al. 2014). Thinning projects may simply improve the water balance of the remaining trees on the landscape (Otero et al. 2010). Differences in water use among species, which include seasonality of transpiration and use of water from different strata (Link et al. 2014) add further complications to forest manipulation intended to increase streamflow.

Large differences in water yield between basins suggest that differences in internal plumbing sets limits on the potential for baseflow increases due to vegetation manipulation or instream treatments. While land management could produce incremental gains in flow that could have positive benefits for aquatic biota, a lack of subsurface storage capacity in the drier basins may limit their response.



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





**Table 1. Summary of streamflow measurement sites, drainage areas, number of flow measurements completed at each site, and the extent of dry reach surveys.**

| Stream | Sub-drainage | Site code | Drainage area (km$^2$) | No. of flow measurements |
|---|---|---|---|---|
| Ancestor Creek | Main | anc1 | 2.58 | 5 |
| | East Fork | ance1 | 0.44 | 4 |
| | Hulse Fork | anch1 | 0.17 | 5 |
| | West Fork | ancw1 | 0.67 | 5 |
| Anderson Creek | Lower | and1 | 1.78 | 5 |
| | Mid | and2 | 1.39 | 3 |
| | Upper | and5 | 0.87 | 3 |
| Baker Creek | Lower | bak1 | 3.98 | 3 |
| | Upper | bak3 | 3.67 | 7 |
| Helen Barnum Creek | Lower | heb1 | 1.66 | 11 |
| | Upper | heb4 | 1.19 | 7 |
| Lost River | Lower | lor1 | 3.52 | 7 |
| | Mid | lor3 | 3.44 | 4 |
| | Upper | lor7 | 3.24 | 5 |
| | Lower North Fork | nlor1 | 1.33 | 8 |
| | Upper North Fork | nlor2 | 1.27 | 5 |
| | Lower South Fork | lors1 | 1.88 | 7 |
| | Upper South Fork | lors3 | 1.22 | 6 |
| Thompson Creek | South Fork | uth1 | 2.28 | 2 |
| | Danny's Creek | dan1 | 1.24 | 2 |
| | Yew Creek | yew | 2.32 | 1[1] |
| Van Arken Creek | Mouth | vaa0 | 5.25 | 0[2] |
| | Lower | vaa3 | 4.81 | 1 |
| | Mid | vaa1 | 3.26 | 1 |
| | Upper | vaa2 | 1.90 | 1 |

[1] Used additional measurement from 10/5/04 (Mattole Salmon Group unpublished data)

5   [2] Used 12 measurements from 2006-2011 (Klein 2009 and Sanctuary Forest unpublished data)





**Table 2. Description of the specific discharge ($Q_{sp}$) metrics used in correlation analysis with the basin variables listed in Table 3..**

| Abbreviation | Number of sites | Definition and details |
|---|---|---|
| $Q_{76}$ | 14 | Tributary $Q_{sp}$ when daily mean Ettersburg Q was 425 l s$^{-1}$, a 76% exceedance flow. Measurements from 10 sites were taken on 1 & 2 November, 2011. Values for the other four sites were interpolated from higher and lower discharges. |
| $Q_{85}$ | 20 | Tributary $Q_{sp}$ when daily mean Ettersburg Q was 272 l s$^{-1}$, an 85% exceedance flow. Measurements from 10 sites were taken on 10 September 2011. Values for the other 10 sites were interpolated from higher and lower discharges. |
| $Q_{96}$ | 18 | Tributary $Q_{sp}$ when daily mean Ettersburg Q was 150 l s$^{-1}$, a 96% exceedance flow. Measurements from 16 sites were taken on 3 & 4 October 2012. Values for the other two sites were interpolated from higher and lower discharges. |



**Table 3. Basin characteristics used in correlation analysis with flow data.**

| Metric | Abbreviation | Definition |
|---|---|---|
| E, N, W, or S facing | EF.frac, NF.frac, WF.frac, SF.frac | Fraction of basin pixels facing 45°-135°, 315°-45°, 225°-315° or 135°-225 respectively |
| Basin eastness | eastness | Measure of basin east-facing (-1 to +1) –mean basin value of sine *(pixel aspect in radians)* (Wilson et al. 2007) |
| Q2 Basin northness | northness | Measure of basin north-facing (-1 to +1) –mean basin value of cosine *(pixel aspect in radians)* (Wilson et al. 2007) |
| Bifurcation ratio | bifur.ratio | Mean (sum of stream channel length order n/sum length order n+1), where n does not equal max order (Price et al. 2011) |
| Channel gradient, 25th, 50th and 75th percentile | CkGradQ25-CkGradQ75 | Calculated by rasterizing mean gradient for each stream segment from NOAA IP layer (Agrawal et al. 2005, Miller 2002) |
| Drainage density | dd.km | Total stream length (km) divided by basin area (km$^2$) |
| First order stream fraction | first.order | Fraction of channel network comprised of 1st order streams, by length |
| Stream gradient <3% | grad.less3 | Fraction of channel network by length with gradient less than three percent |
| Stream length, sum | str.length | Total stream length in basin (m) |
| Distance to coast | dist.coast | Shortest path distance (km) from basin centroid to ocean |
| Precipitation mean, April, May, June, July, August, Sept. or Oct. | M4norm - M10norm | Basin precipitation by month, 1981-2010 average, PRISM data (mm) |
| June, July, or August Temperature, maximum | M6tmax - M8tmax | Basin average monthly maximum temperature (°C), 1981-2010, PRISM data |
| June, July or August Temperature, mean | M6tmean – M8tmean | Basin average monthly mean temperature (°C), 1981-2010, PRISM data |
| Debris slide slopes | dss.frac | Proportion of basin mapped as slopes sculpted by debris slides or debris flows (CGS 2003, Davenport et al. 2002) |
| Quaternary alluvium | qoal.frac | Proportion of basin mantled by Holocene or Pleistocene alluvium (CGS 2003, Davenport et al. 2002) |
| Area to perimeter ratio | a.p.ratio | Basin area divided by basin perimeter |
| Basin area | Area.km | Basin area (km$^2$) |
| Max. flowpath length/basin area | max.fpl.area | MaxFPL (km) divided by basin area (km$^2$) |
| Maximum flowpath length | MaxFPL | Maximum flowpath distance from basin divide to outlet. Includes both overland flow and the stream network. |
| Basin perimeter | Perim.m | Basin perimeter (km) |
| Riparian buffering ratio, 25th, 50th and 75th percentile | rh2Q25, rh2Q50, rh2, Q75 | Ratio of riparian area/hillslope area (m$^2$) draining to each stream pixel, calculated for each stream side (Grabs et al. 2010). Riparian area elevations defined as pixels <2m above stream. |
| Riparian area/stream length ratio | rip.per.sl | All cells <1m in height above stream cells, divided by basin stream length |
| Riparian area as fraction of basin | rip1.frac | All cells <1m in height above stream cells, as fraction of basin |
| Valley width, median | VW.q50 | Valley width from IP data layer (Agrawal et al. 2005) estimated as the minimum length of a transect at 2.5 times bankfull depth above channel (Miller 2002). |
| Valley width index, median | VWI.q50 | From IP data layer. Valley width divided by active channel width, as estimated from drainage area. |
| Valley area fraction of basin | val.area.frac | Calculated "valley area" by multiplying stream segment valley width |





| Metric | Abbreviation | Definition |
|---|---|---|
| | | by segment length. |
| Valley area /meter stream length | val.per.sl | Total basin valley area divided by total stream length. |
| Elevation, 25th, 50th, 75th percentile and maximum | elev.q25-elev.q75 and elev.max | Calculated from DEM, meters above sea level |
| Flow Path Gradient, median | FPG.q50 | Gradient along un-channelized flowpath from each pixel to the stream network |
| Flow Path Length, median Flow Path Length, maximum | FPL.q50 FPL.max | Distance along un-channelized flowpath from each pixel to the stream network |
| Basin Flow Path Length/Gradient | FPLG.q50 | Ratio of basin median flow path length to basin median flow path gradient |
| Basin relief | relief | Basin maximum elevation - minimum elevation |
| Basin relief ratio | relief.ratio | Relief divided by maximum flowpathdistance from drainage divide to outlet |
| Basin relative relief | relief.rel | Relief divided by basin perimeter |
| Sub-catchment area, 25th, 50th, 75th percentile | sc.area.q25, sc.area.q50, sc.area.q75 | The catchment areas of all stream pixels in the basin. Lower values indicate a more dendritic channel network. |
| Basin slope, median, maximum, standard deviation | slope.q50, slope.max, slope.sd | Slope value in degrees within basin |
| Topographic wetness index, median | TWI.q50 | $ln(specific\ catchment\ area/\ (tan(slope))$. Greater TWI value predicts greater potential for saturated conditions, e.g. at base of slope where contributing area is large (Beven and Kirkby 1979) |
| Convergence index, mean, 25th percentile, median | ci.mean, ci.q25, ci.q50 | Measure of land surface convergence or divergence, calculated using pixel aspect. Negative values correspond to convergent conditions, positive to divergent (Olaya and Conrad 2009). |
| Upslope accumulated area, 25th, 50th, 75th, 90th percentile, and maximum | UAAQ25 - UAAQ90, UAAMAX | Hillslope area ($m^2$) draining to each stream pixel (via un-channelized flow, not including contribution from upstream), calculated separately for each stream side (Grabs et al. 2010, Jencso et al. 2009, McGlynn and Seibert 2003) |
| Topographic position index, mean, median | TPI.mean, TPI.q50 | Measure of a elevation relative to surroundings. Positive TPI value indicates higher than surrounding areas, (Wilson et al. 2007) |
| Tree canopy density | canopy.density | Area-weighted mean of total tree canopy cover, from "DEN_TOTAL" field in CalVEG (CDF 2005) |
| Coniferous, Hardwood or Mixed forest type | Conifer, Hardw or MixTreeFrac | Proportion of basin mapped as conifer, hardwood or mixed conifer/hardwood dominated (WHR field in CalVeg) (CDF 2005) |
| Lower ET vegetation | low.et.frac | Fraction of basin area with largest tree size class (30-39"dbh), and barren, grass, and sapling classes from CalVeg |
| Mean tree diameter in basin | tree.dia.mean | From areal extent of size classes in CalVeg "NWSIZE" field |
| Parcel density | parcel.density | Unique parcel number count divided by basin area ($km^2$) |





**Table 4. Spearman's correlation coefficients (*rho*) for basin variables. Values are in bold where p<0.01.**

| Metric | Abbreviation | Q76 (n=14) | Q85 (n=20) | Q96 (n=18) |
|---|---|---|---|---|
| **Aspect** | | | | |
| East facing aspect | EF.frac | 0.36 | 0.03 | 0.05 |
| North facing aspect | NF.frac | 0.10 | 0.09 | -0.07 |
| West facing aspect | WF.frac | -0.45 | -0.25 | -0.25 |
| South facing aspect | SF.frac | -0.17 | 0.39 | 0.50 |
| Basin Eastness | eastness | 0.34 | 0.12 | 0.10 |
| Basin Northness | northness | 0.27 | -0.15 | -0.28 |
| **Channel network** | | | | |
| Bifurication ratio | bifur.ratio | -0.35 | -0.33 | -0.43 |
| Channel gradient, median | CkGradQ50 | 0.56 | **0.84** | **0.82** |
| Drainage density | dd.km | **0.78** | **0.71** | 0.54 |
| First order stream fraction | first.order | 0.13 | 0.31 | 0.41 |
| Stream gradient <3% | grad.less3 | **-0.72** | **-0.76** | **-0.82** |
| Stream length, sum | str.length | -0.21 | -0.44 | **-0.63** |
| **Geology** | | | | |
| Debris slide slopes | dss.frac | 0.20 | 0.11 | -0.10 |
| Quaternary alluvium | qoal.frac | **-0.81** | **-0.84** | **-0.84** |
| **Basin Morphometry** | | | | |
| Area to perimeter ratio | a.p.ratio | -0.20 | -0.12 | -0.27 |
| Basin area | Area.km | -0.29 | -0.51 | **-0.66** |
| Maximum flowpath length/basin area | max.fpl.area | 0.08 | 0.25 | 0.46 |
| Maximum flowpath length | MaxFPL | -0.55 | **-0.70** | **-0.75** |
| Basin perimeter | Perim.m | -0.31 | **-0.65** | **-0.81** |
| **Riparian/valley character** | | | | |
| Riparian buffering ratio, 25th percentile | rh2Q25 | -0.36 | **-0.77** | **-0.80** |
| Riparian buffering ratio, median | rh2Q50 | -0.57 | **-0.59** | -0.46 |
| Riparian buffering ratio, 75th percentile | rh2Q75 | -0.65 | **-0.71** | -0.42 |
| Riparian area/stream length ratio | rip.per.sl | **-0.71** | **-0.81** | **-0.81** |
| Riparian area as fraction of basin | rip1.frac | -0.36 | **-0.58** | **-0.68** |
| Valley width, median | VW.q50 | **-0.76** | **-0.88** | **-0.72** |
| Valley width index, median | VWI.q50 | **-0.77** | **-0.88** | **-0.81** |
| Valley area fraction of basin | val.area.frac | -0.63 | **-0.79** | **-0.66** |
| Valley area per meter of stream length | val.per.sl | **-0.85** | **-0.85** | **-0.61** |




| Basin topography | | | | |
|---|---|---|---|---|
| Elevation, median | elev.q50 | **0.75** | **0.83** | **0.83** |
| Elevation, maximum | elev.max | 0.47 | 0.36 | 0.31 |
| Flow Path Gradient, median | FPG.q50 | 0.63 | **0.85** | **0.76** |
| Flow Path Length, median | FPL.q50 | **-0.75** | **-0.68** | -0.46 |
| Flow Path Length, maximum | FPL.max | -0.38 | -0.15 | -0.01 |
| Basin Flow Path Length/Gradient | FPLG.q50 | **-0.77** | **-0.81** | **-0.63** |
| Basin relief | relief | -0.09 | -0.15 | -0.09 |
| Basin relief ratio | relief.ratio | 0.66 | **0.80** | **0.82** |
| Basin relative relief | relief.rel | 0.49 | **0.67** | **0.86** |
| Sub-catchment area, 25th percentile | sc.area.q25 | -0.57 | -0.57 | **-0.67** |
| Sub-catchment area, median | sc.area.q50 | -0.65 | **-0.80** | **-0.80** |
| Sub-catchment area, 75th percentile | sc.area.q75 | -0.62 | **-0.84** | **-0.94** |
| Basin slope, median | slope.q50 | **0.68** | **0.81** | 0.72 |
| Basin slope, maximum | slope.max | 0.15 | 0.10 | -0.21 |
| Basin slope, standard deviation | slope.sd | 0.37 | 0.00 | -0.12 |
| Topographic wetness index, median | TWI.q50 | -0.62 | **-0.84** | **-0.74** |
| Upslope accumulated area, 25th percentile | UAAQ25 | 0.24 | 0.14 | -0.17 |
| Upslope accumulated area, median | UAAQ50 | -0.62 | **-0.79** | **-0.80** |
| Upslope accumulated area, 75th percentile | UAAQ75 | **-0.84** | **-0.73** | -0.58 |
| Upslope accumulated area, 90th percentile | UAAQ90 | **-0.78** | **-0.69** | -0.50 |
| Upslope accumulated area, maximum | UAAMAX | **-0.79** | **-0.80** | **-0.63** |
| Convergence index, mean | ci.mean | -0.01 | 0.42 | 0.47 |
| Convergence index, 25th percentile | ci.q25 | -0.05 | -0.26 | **-0.79** |
| Convergence index, median | ci.q50 | 0.35 | 0.43 | 0.33 |
| Topographic position index, mean | TPI.mean | -0.07 | 0.33 | 0.14 |
| Topographic position index, median | TPI.q50 | 0.67 | **0.78** | **0.61** |
| **Vegetation** | | | | |
| Tree canopy density | canopy.density | **0.79** | **0.70** | 0.57 |
| Coniferous forest type | ConiferFrac | -0.25 | **-0.64** | -0.56 |
| Hardwood forest type | HardwFrac | 0.47 | **0.65** | **0.66** |
| Mixed forest type | MixTreeFrac | -0.58 | -0.04 | 0.06 |
| Lower ET vegetation | low.et.frac | -0.49 | -0.53 | -0.43 |
| Mean tree diameter in basin | tree.dia.mean | 0.17 | -0.18 | -0.42 |
| **Human water use** | | | | |
| Parcel density | parcel.density | 0.60 | **0.75** | **0.66** |





**Table 5. Spearman's correlation coefficients (*rho*) for basin normal (1981-2010) precipitation and temperature, and flow metrics. Values are in bold where *p<0.01*.**

| Metric | Abbreviation | Q76 (n=14) | Q85 (n=20) | Q96 (n=18) |
|---|---|---|---|---|
| Distance to coast | dist.coast | **-0.79** | **-0.71** | -0.54 |
| April Precipitation, mean | M4norm | 0.39 | **0.69** | **0.75** |
| May Precipitation, mean | M5norm | 0.64 | **0.84** | **0.72** |
| June Precipitation, mean | M6norm | **0.84** | **0.79** | 0.60 |
| July Precipitation, mean | M7norm | 0.51 | -0.05 | -0.14 |
| August Precipitation, mean | M8norm | **0.80** | **0.75** | **0.60** |
| September Precipitation, mean | M9norm | 0.67 | **0.84** | **0.78** |
| October Precipitation, mean | M10norm | 0.39 | **0.69** | **0.74** |
| June Temperature, maximum | M6tmax | **-0.79** | **-0.75** | **-0.64** |
| July Temperature, maximum | M7tmax | -0.60 | -0.56 | -0.36 |
| August Temperature, maximum | M8tmax | **-0.80** | **-0.75** | **-0.64** |
| June Temperature, mean | M6tmean | 0.31 | 0.34 | 0.25 |
| July Temperature, mean | M7tmean | 0.59 | **0.79** | **0.82** |
| August Temperature, mean | M8tmean | **0.93** | **0.79** | **0.66** |





**Table 6. Spearman's correlation coefficients (*rho*) for mean basin monthly precipitation totals during the study period and flow metrics. Precipitation data from 2011 was used with $Q_{76}$, $Q_{85}$; and 2012 data with $Q_{96}$ values, since for most sites $Q_{sp}$ data were collected in these respective years. Values are in bold where $p<0.01$.**

| Month | $Q_{76}$ (n=14) | $Q_{85}$ (n=20) | $Q_{96}$ (n=18) |
|---|---|---|---|
| April | **-0.82** | **-0.73** | **0.75** |
| May | 0.43 | **0.75** | **0.67** |
| June | **0.83** | **0.76** | 0.57 |
| July | 0.40 | -0.25 | -0.50 |
| August | | | 0.34 |
| September | 0.49 | **0.79** | |
| October | 0.45 | | |





**Table 7. Variables with high loadings, and the proportion of variance explained by components from PCA.**

All variables from Table 4 (n=47) except weather/climate variables

| Component | Strong Positive Loadings | Strong Negative Loadings | Proportion of variance |
|---|---|---|---|
| 1 | VW.q50, VWI.q50, qoal.frac, TWI.q50, rip.per.sl, FPLG.q50, val.per.sl, grad.less.3, sc.area.q50, UAAQ50 | Slope.q50, FPG.q50, elev.q50, CkGradQ50, canopy.density, relief.rel, dd.km, TPI.q50 | 0.6324 |
| 2 | SF.frac, MixTreeFrac, max.fpl.area, FPL.q50 | Northness, NF.frac, ConiferFrac, tree.dia.mean, Perim.m, dd.km | 0.1798 |
| 3 | Rip1.frac, grad.less3, HardwFrac, TPI.q50, elev.q50, tree.dia.mean, rip.per.sl, UAAq50 | First.order, FPL.max, relief, CkGradQ50, dist.coast, val.area.frac | 0.1441 |

Weather/climate, aspect, and vegetation variables (n=35)

| Component | Strong Positive Loadings | Strong Negative Loadings | Proportion of variance |
|---|---|---|---|
| 1 | M6.2012, M6norm, M9norm, M6.2011,M5norm, M5.2011, M9.2011, M5.2012, M8tmean, M8norm, Canopy.density | M4.2011, M8tmax, M6tmax | 0.5852 |
| 2 | M7tmax, M4.2011, M4norm, M10norm, M4.2012, M8tmax, M6tmax | M7norm, M7.2011, M7.2012, M8.2012, tree.dia.mean, HardwFrac | 0.3028 |







**Figure 1.** Map of the study area in the Southern subbasin of the Mattole River watershed, Humboldt and Mendocino counties, California. Inset shows general location in region.





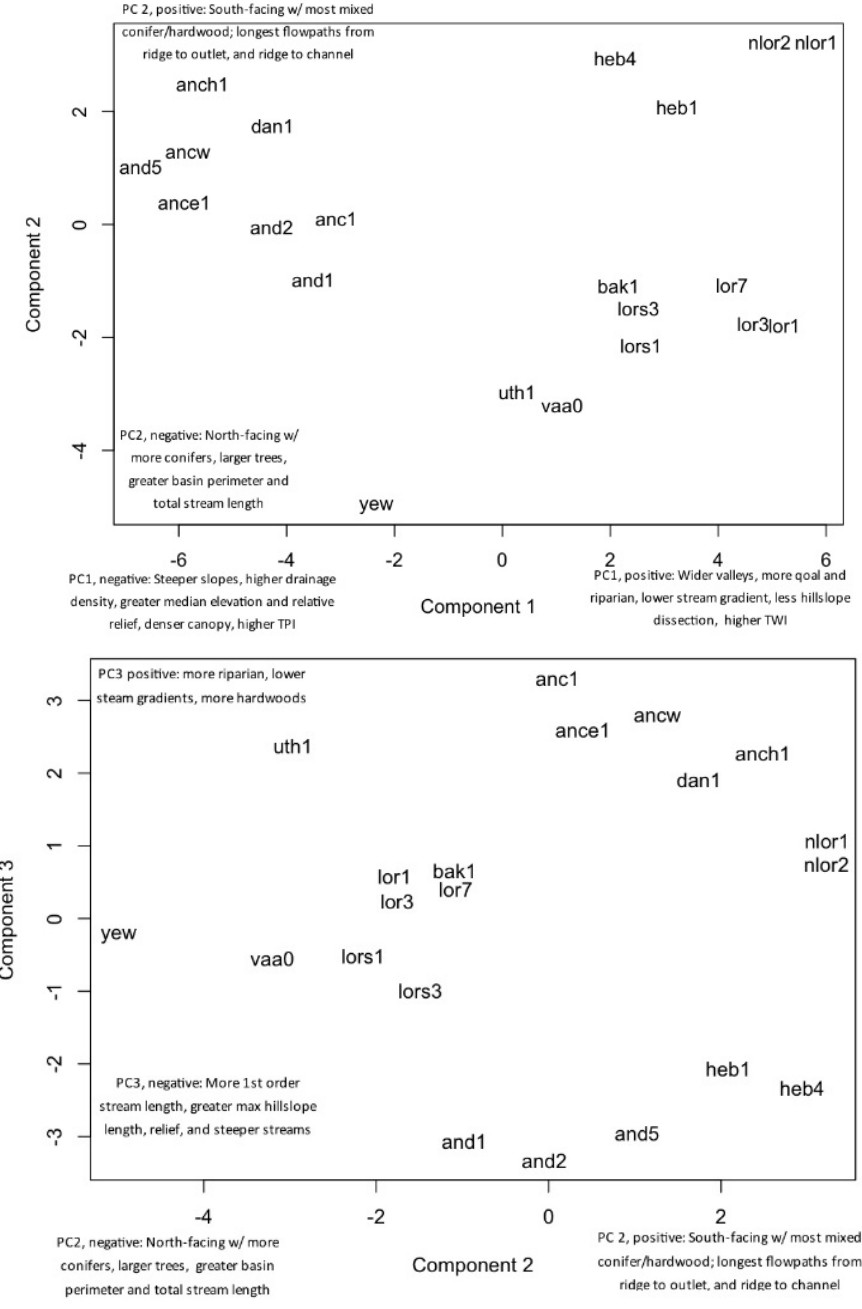

Figure 2. Principle component analysis for climate and topographic variables influencing baseflow. Abbreviations for subcatchment names are in Table 1. Annotation describes variables with highest loadings for positive and negative position along axis of each component. Above: Components 1 and 2 from PCA using all variables from Table 3, except for weather/climate variables. Below: Components 2 and 3 from PCA using all variables from Table 3, except for weather/climate variables.

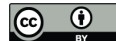



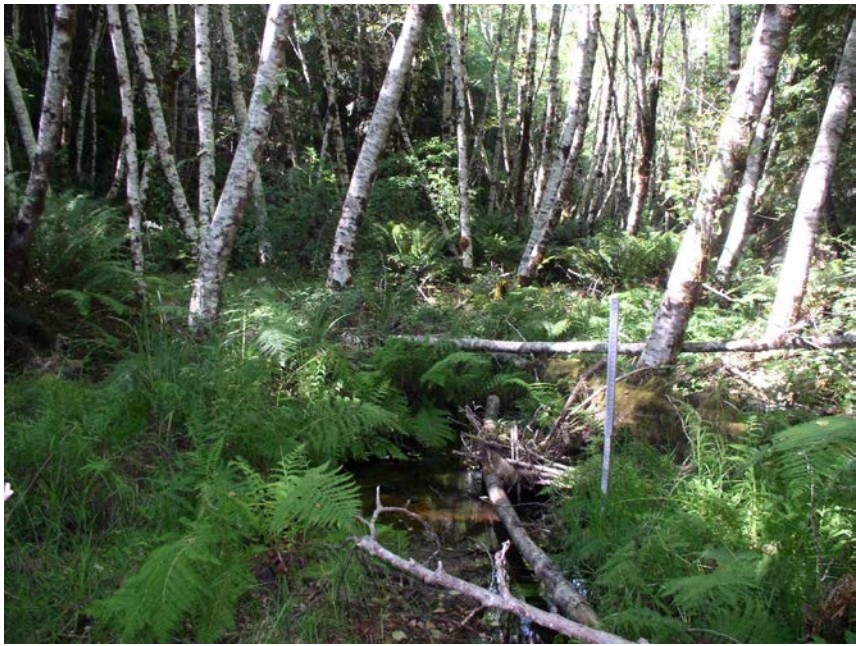

a)

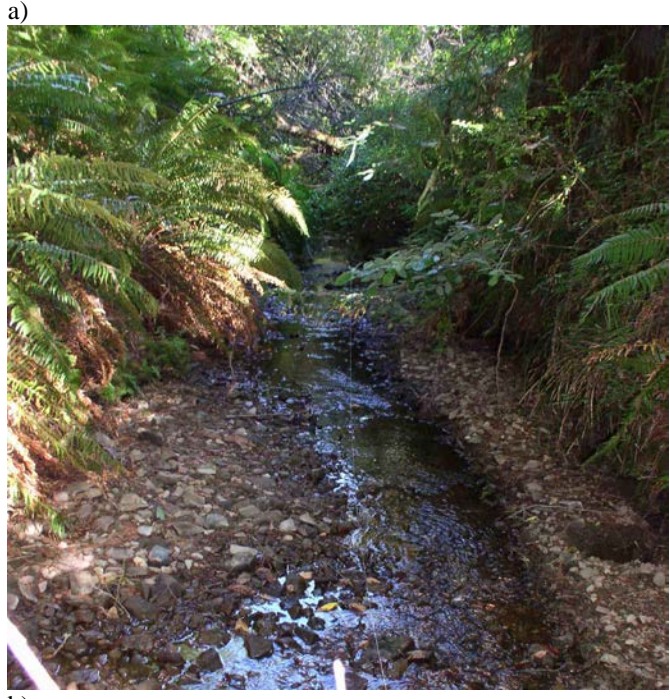

b)

5   **Figure 3 South Fork of the Lost River. a) Alder-dominated riparian area with extensive connected floodplain, typical of the riparian area within the Upper South Fork. b) Semi-confined channel with limited floodplain connectivity between lower and upper flow measurement sites. Limited floodplain connectivity is typical of the South Fork of Lost River downstream of the Upper South Fork site, and the main fork of Lost River.**



**Figure 4. Slope, GIS-derived channel network, and extent of the valley floor less than one meter above channel elevation in Ancestor Creek and Lost River, catchments with higher and lower baseflows, respectively. Subcatchment names given in Table 1.**