# Peer review of "Spatial and Temporal Variability in Baseflow in the Mattole River Headwaters, California, USA"

_Hydrology and Earth System Sciences, 2016_

## Referee Comment (RC1) · Anonymous Referee #1 · 18 Aug 2016

General comments

This manuscript presents an exercise made for analyzing the relationships between several catchment descriptors and baseflow rates a set of small headwater catchments. The subject is of interest to the current subjects of catchment hydrological processes, the manuscript is adequately written and presented, the data are original and the outcomes are basically supported by the data analyses. Nevertheless, the statistical design of the exercise is notably weak, because the authors analyze the role of over 60 catchment descriptors on just 3 baseflow variables without sufficiently taking into account the redundancy among most of the descriptors. Furthermore, most of the baseflow variables were obtained using linear regressions without estimating the

uncertainty associated to these interpolation methods. Some of the outcomes are consistent and of value in the light of the recent findings in the subject, but some others appear to be due to the correlation among diverse catchment descriptors, so statistical correlations do not necessarily mean causal relationships. Before interpreting the results, the authors should show a correlation table between the catchment descriptors and use the Principal Component Analysis made for mapping these descriptors. The interpretation of the causal relationships must be done taking into account more the factorial axes than the separate variables. This means that most of the results and discussion sections must be rewritten accordingly.

Specific and technical comments

- Page 2 line 21 and elsewhere: Smakhatin should be Smakhtin

- Page 3 line 7: a recently published work made using large scale data shows that high gradient catchments have less young streamflow (Jasechko, S., Kirchner, J.W., Welker, J.M. and McDonnell, J.J., 2016. Substantial proportion of global streamflow less than three months old. Nature Geoscience, 9: 126-123.)

- Page 4 line 23: mm are typically used instead of cm.

-Page 4 lines 27-29: Genus should be fully stated along with species.

- Page 5 line 1: some more information on the age of the rocks as well on their degree of tectonic and metamorphic modifications would be of value taking into account the subject of the paper.

- page 6 lines 9-11: the two sentences may be deleted.

-page 6, lines 11-18: the uncertainty associated to these interpolation methods should be analyzed.

- page 7, lines 3-7: If climatic data for the catchments are derived from their topographic characteristics, both the original and derived variables should not be equally analyzed

for their role on baseflows. Subsequently, cause-effect relationships should be used instead of statistical ones.

-page 7, line 32: explain how these cooler and humid episodes affected the recession flows.

- page 10, lines 12-16. Canopy density seems to be strongly correlated to topographic gradient, so this may not be a causal relationship.

- page 16, lines 21 to 28: these are not properly conclusions, because this subject was not analyzed before. Please, move the paragraph to the discussion section with a new header such as '4.5. Implications', or change the name of this section into 'concluding remarks'. Line 21: "unit/area baseflow yield"

Table 1: "the extent of dry reach surveys" is not shown in the table

Figure 1: state the names of the States in the inset

Figure 2: This graph is not necessary given the written explanations. Instead, a map of the catchment variables would be necessary. The fraction of the correlation explained by the axes should be stated in the caption.

---

## Referee Comment (RC2) · Anonymous Referee #2 · 20 Aug 2016

General comments:

The paper deals with the interesting topic of relating basin characteristics to base flow rates for a small scale catchments, and clearly testifies from a lot of work and efforts that have been done to compile it. In General the manuscript is adequately written, and the methods are well described. Discussion and conclusions are well-supported by the results. The figures and tables are nicely presented with sufficient legends and captions, though some tables and figures should be modified. However, the PCA statistical analysis are not well described and should be modified. First a coloration matrix should be constructed for all the variables to test the collinearity (New table should be added). After removing the highly correlated variables, a PCA analysis could

be performed. The Author also should test the rotation of the PCA components in order to maximize the variance among the variables under each component.

Specific remarks

• Page6, Line 10: stream not "stem". • Page 7, Line 17: PCA was based on standardized variables or original variables? • Page 11, Line 6: "Principal components analysis was run on two subsets of the variables" • A table should be added for the two runs with the eigenvalue and the variance of each principal component, also the loading of each variable. • Page 11, Line 11: Which criteria has been chosen to detect the high loading factor? • Why didn't you try to rotate the PCA components to maximize the variance? • Page 10, Line 16: "none of the vegetation metrics showed particularly strong correlations with the flow metrics" A recent study was conducted to estimate the controlling factors of base flow using PCA analysis, and vegetation was found to be the first dominating factor for spatial variation of base flow. (Zomlot, Z., Verbeiren, B., Huysmans, M., Batelaan, O. (2015). Spatial distribution of groundwater recharge and base flow: assessment of controlling factors. Journal of Hydrology: Regional Studies, 4(B), 349-368.) • Figure2: The figure looks a bit crowded, clusters should be marked with colours or dashed lines to be clearly seen. The Yew catchment in Figure 2: seems to be an outlier?

————————————————————

---

## Author Comment (AC1) · 4 Oct 2016

We thank the reviewer for their detailed comments. Here is our reply.

Anonymous Referee #1 comments

"Nevertheless, the statistical design of the exercise is notably weak, because the authors analyze the role of over 60 catchment descriptors on just 3 baseflow variables without sufficiently taking into account the redundancy among most of the descriptors... Before interpreting the results, the authors should show a correlation table between the catchment descriptors and use the Principal Component Analysis made for mapping these descriptors. The interpretation of the causal relationships

must be done taking into account more the factorial axes than the separate variables. This means that most of the results and discussion sections must be rewritten accordingly."

Response: We agree that the statistical design of the analysis could be strengthened. We intend to examine the correlation matrix and remove highly correlated catchment descriptors, such that no two remaining catchment variables have a correlation coefficient >0.8. Then, conduct Principal Components Analysis using all of the remaining variables. We will rewrite the results and discussion based on the resulting composition of the principal components with eigenvalues greater than one, and the loadings on the variables within those components.

"Furthermore, most of the baseflow variables were obtained using linear regressions without estimating the uncertainty associated to these interpolation methods" – and - page 6, lines 11-18: "the uncertainty associated to these interpolation methods should be analyzed."

Response: The majority of baseflow variables (36 out of 52) were obtained from direct measurements of streamflow (see Table 2 in the manuscript). In the case of the 16 values that were estimated, using this method to obtain values for the baseflow variables is similar to the use of a stage/discharge rating curve to obtain streamflow values from periods when no flow measurement was made. No uncertainty is reported in those cases, another method of estimating "missing" streamflow values. In this case we simply use discharge at the downstream index gaging station, instead of stage.

As we state in the article, the interpolation was only made within the range of measured tributary values. – we did not interpolate up or down beyond the range of measured values, which would potentially introduce substantial error into estimations since streamflow recession is not always a linear function.

At sites where there appeared to be an inflection point in the recession of tributary flows plotted against the index gage, we used only the measurements taken nearest

the desired value on the Ettersburg hydrograph to generate the predictive equation, to avoid fitting a straight line across that inflection point in the recession curve.

Nevertheless, we would be happy to report the error associated with each estimated value. R2 values ranged from 0.89 to 0.99.

"- Page 5 line 1: some more information on the age of the rocks as well on their degree of tectonic and metamorphic modifications would be of value taking into account the subject of the paper."

Response: We will rewrite the 1st sentence on page 5 to read: The underlying geology is highly folded and variably sheared sandstone and argillite, classified as Late Cretaceous to Pliocene-aged rocks of the Coastal Belt of the Franciscan Complex (Davenport et al. 2002).

"- page 7, lines 3-7: If climatic data for the catchments are derived from their topographic characteristics, both the original and derived variables should not be equally analyzed for their role on baseflows. Subsequently, cause-effect relationships should be used instead of statistical ones."

Response: When removing highly correlated catchment descriptors as described previously, we will also remove the physiographic variables used to derive precipitation and temperature records.

"-page 7, line 32: explain how these cooler and humid episodes affected the recession flows."

Response: In general, cooler and humid episodes seemed to decrease the rate of recession or caused flows to increase slightly in streams with surface flows throughout most of the drainage network, but appeared to have little impact in streams with long stretches of subsurface or intermittent flow. However, our synoptic measurements didn't have the necessary temporal resolution to describe this phenomenon with a high degree of certainty.

Table 1: "the extent of dry reach surveys" is not shown in the table Response: We will delete this from the Table description.

"Figure 2: This graph is not necessary given the written explanations. Instead, a map of the catchment variables would be necessary. The fraction of the correlation explained by the axes should be stated in the caption."

Response: This figure will be revised accordingly after re-doing the statistical analysis as suggested by reviewers.

---

## Author Comment (AC2) · 4 Oct 2016

We appreciate the time and effort of the reviewer. Our responses are below. Anonymous Referee #2 comments NQ 10/3/2016

"However, the PCA statistical analysis are not well described and should be modified. First a coloration matrix should be constructed for all the variables to test the collinearity (New table should be added). After removing the highly correlated variables, a PCA analysis could be performed. The Author also should test the rotation of the PCA components in order to maximize the variance among the variables under each component.

Response: We agree that the statistical design of the analysis could be strengthened.

We intend to examine the correlation matrix and remove highly correlated catchment descriptors, such that no two remaining catchment variables have a correlation coefficient >0.8. Then, conduct Principal Components Analysis, including varimax rotation, using all of the remaining variables. We will rewrite the results and discussion based on the resulting composition of the principal components with eigenvalues greater than one, and the loadings on the variables within those components.

"Page 7, Line 17: PCA was based on standardized variables or original variables?"

Response: Variables were standardized within the function PcaHubert in R, using "Scale=True".

"Page 11, Line 6: "Principal components analysis was run on two subsets of the variables. A table should be added for the two runs with the eigenvalue and the variance of each principal component, also the loading of each variable."

Response: After re-doing the PCA with fewer highly correlated variables, as described above, there should no longer be any need to break the variables into two subsets. We will add a table reporting the eigenvalues and variable loadings.

Page 11, Line 11: Which criteria has been chosen to detect the high loading factor? Why didn't you try to rotate the PCA components to maximize the variance?

Response: No specific criteria was set for a "high" loading factor for variables. The variable loadings were examined, and those with the greatest absolute values were described as having strong or heavy loadings. Varimax rotation should have been performed.

Line 16: "none of the vegetation metrics showed particularly strong correlations with the flow metrics" A recent study was conducted to estimate the controlling factors of base flow using PCA analysis, and vegetation was found to be the first dominating factor for spatial variation of base flow. (Zomlot, Z., Verbeiren, B., Huysmans, M., Batelaan, O. (2015). Spatial distribution of groundwater recharge and base flow: assessment of controlling factors. Journal of Hydrology: Regional Studies, 4(B), 349-368.)

Response: The relative homogeneity in the vegetation within our study area, in contrast to the study cited above, may have been partially responsible for the lack of correlation between flow and vegetation.

"Figure2: The figure looks a bit crowded, clusters should be marked with colours or dashed lines to be clearly seen. The Yew catchment in Figure 2: seems to be an outlier?"

Response: This figure will be revised accordingly after re-doing the statistical analysis as suggested by reviewers.

The Yew catchment does seem to be an outlier. It differs in both aspect and vegetation from most of the other catchments, and is notable for having the most old-growth forest of any of the catchments in this study. PCA conducted with fewer highly correlated variables may help elucidate how this catchment differs, or doesn't, from the others with relatively higher flows.

---

## Editor Comment (EC1) · J. Seibert (Editor) · 5 Oct 2016

Dear Reviewers and authors, thanks for your valuable comments and responses. While the former highlight that improvements are needed, especially when it comes to the statistical analyses, the latter indicates that the authors are well-aware what is needed to improve the manuscript. While major revisions are needed, I am optimistic that these will largely improve the manuscript.

Best regards, Jan Seibert
* * *